# Multi-Source T-S Target Recognition via an Intuitionistic Fuzzy Method

**Chuyun Zhang** [1,2], **Weixin Xie** [1,2], **Yanshan Li** [1,2] **and Zongxiang Liu** [1,2,*]

1    Guangdong Key Laboratory of Intelligent Information Processing, Shenzhen University, Shenzhen 518060, China; zhangchuyun2019@email.szu.edu.cn (C.Z.)
2    College of Electronics and Information Engineering, Shenzhen University, Shenzhen 518060, China
*    Correspondence: liuzx@szu.edu.cn; Tel.: +86-755-26732055

**Abstract:** To realize aerial target recognition in a complex environment, we propose a multi-source Takagi–Sugeno (T-S) intuitionistic fuzzy rules method (MTS-IFRM). In the proposed method, to improve the robustness of the training process of the model, the features of the aerial targets are classified as the input results of the corresponding T-S target recognition model. The intuitionistic fuzzy approach and ridge regression method are used in the consequent identification, which constructs a regression model. To train the premise parameter and reduce the influence of data noise, novel intuitionistic fuzzy C-regression clustering based on dynamic optimization is proposed. Moreover, a modified adaptive weight algorithm is presented to obtain the final outputs, which improves the classification accuracy of the corresponding model. Finally, the experimental results show that the proposed method can effectively recognize the typical aerial targets in error-free and error-prone environments, and that its performance is better than other methods proposed for aerial target recognition.

**Keywords:** target recognition; T-S intuitionistic fuzzy rules; ridge regression; adaptive weight

## 1. Introduction

The complexity of the battlefield environment is enhanced significantly by high-tech equipment, which has introduced great difficulties to the acquisition of target information. As the battlefield expands to the five-dimensional space of sea, land, air, sky, and electromagnetics, the collection of target information will not only be affected by the accuracy and stability of sensor equipment, the influences of the climate environment, and the complex electromagnetic field environment, but also by other factors that lead to deviations or even errors in the collected target information. In addition, there will be interference and confusing equipment intentionally released by the enemy, which increases the uncertainty of the observation of the target. Therefore, it is difficult for a single information source to obtain accurate and complete intelligence information in such a complex environment, and also meet the requirements of actual aerial combat.

With the development of multi-source detection technology, a structure able to track multiple targets and realize target recognition is essential to a multi-sensor data fusion system. Information fusion can recognize a target from multiple dimensions and multiple directions, which data can then be comprehensively processed with the complementarity and redundancy of information, to eliminate the influence of inaccuracy and incompleteness of information obtained from a single information source. Moreover, multi-feature fusion processing is designed to obtain more accurate target features by data fusion of two or more sensors, thus breaking the limits of single-sensor detection, in which equipment generally collects the information of only one feature within a corresponding sensing range [1]. Target features obtained by different sensors are imprecise and conflict with the influence of complex environments, interference signals and so on; for example, impulsive noise may

cause the collected data to deviate from the original range, leading to the drawing of the wrong conclusions in the target recognition system. Therefore, muti-feature fusion and improving the interpretability of target recognition are particularly important.

### 1.1. Literature Review and Motivation

For the recognition system, a series of methods have been presented, such as the Dempster–Shafer (D-S) [2–4], fuzzy set [5–7], probability statistics [8–10], the gray system [11,12], rough sets [13–15], and fractal theory [16–18]. D-S evidence theory, a general framework for information fusion, is used to combine multi-level information from multi-source environments for reasoning and dealing with uncertainty, imprecision, and incompletion [19,20]. Therefore, extended evidence theories have been well established in information fusion [21], decision analysis [22], risk assessment [23,24], pattern recognition [25], and other fields. However, traditional evidence theory has low accuracy because of the problems of constructing a basic probability assignment (BPA) and conflict management.

With regard to the framework of the BPA, some modeling approaches have been provided. Moreover, Dempster's combination method is performed to transform the BPA into probability distribution, the quality of the BPA in evidence theory will determine whether the recognition result is reasonable. Yin et al. [26] proposed a measurement model to achieve uncertainty management of the BPA via the processing of negation and the links between uncertain data and entropy. Jiang et al. [27] constructed a correlation coefficient to describe the non-intersection and the distinctions between the focal elements. Wang et al. [28] proposed a belief divergence measurement that presented the correlation of various kinds of subsets with a belief function and an appropriate probability distribution. Kaur et al. [8] processed nonnegative and symmetric divergence measures for BPA. Hu et al. [9] proposed the cross-information to change the comprehensive BPA. However, an algorithm based on decision-level data fusion needs high data preprocessing and the decision-making methods are short of general structure after obtaining the characterized distributions of basis reliability.

When coping with highly conflicting evidence, D-S evidence theory may lead to counter-intuitionistic recognition. Therefore, many methods have been proposed including Yager's combination rules method [29], Murphy's arithmetically average model of bodies of evidence [30], Li's trust-based method [31], and so on. Target recognition methods based on fuzzy set theory only need a small amount of prior knowledge to achieve more efficient and accurate recognition. Wang [32] proposed the intuitionistic fuzzy dynamic Bayesian network to transfer the outputs of intuitionistic fuzzy rules into probability. Jiang [7] established a hybrid decision-making fuzzy rough and hesitant sets model and developed a machine learning mechanism to construct the relative loss functions. Guo [33] proposed the recognition structure of UAVs based on a recurrent convolutional strategy, which influenced the degrees of super-resolution realization by setting the numbers of cycles and iterations with changes in the blur degree. Moreover, intuitionistic fuzzy sets (IFS) can conquer the inaccuracy and limitations of traditional fuzzy sets for solving specific information and eliminate the bottleneck that Bayesian models excessively rely on. Lei [34] proposed an intuitionistic fuzzy reasoning (IFR) framework to obtain the membership and non-membership degrees of the property variables of a recognition model. Dolgiy [35] combined the D-S method and Takagi–Sugeno (T-S) fuzzy system to develop the empirical process of an expert system of probability estimates based on subjective preferences of the description of typical sensors. Therefore, a novel hybrid T-S and intuitionistic fuzzy inference system are applied to target recognition in our method.

### 1.2. Our Contributions

In this paper, a novel MTS-IFRM is proposed for high-performance multi-target recognition in error-free and error-prone environments. The main novelties of our method include:

- Improving the robustness of the training process of the model: the features of the aerial targets are classified as inputs to the corresponding T-S target recognition model, so that features are divided into multi-level features with the target properties;
- In the T-S model algorithm, the study of premise and consequence parameter identification has been the key question. We apply an intuitionistic fuzzy C-means method based on the dynamic particle swarm optimization (DPSO) algorithm and the ridge regression model to identify the premise and consequence parameter of the T-S intuitionistic fuzzy model, respectively, which better realizes the parametric identification of the model;
- High classification accuracy can be guaranteed in error-free and error-prone environments. The adaptive weight algorithm reduces the weight corresponding to the model with a low degree of discrimination and increases the weight corresponding to the model with a high degree of discrimination, which is better distinguished from the input features.

### 1.3. Organization of the Article

The organization of the method is described as follows: The fuzzy target recognition model is given in Section 2. Model construction and parameter identification are presented in Section 3. The simulation results and an analysis with comparable methods are given in Section 4. Finally, the conclusions are organized in Section 5. The meanings of notation in the article are listed in Table 1.

**Table 1.** Notation list.

| Notation | Meaning of the Notation | Notation | Meaning of the Notation |
|---|---|---|---|
| $\Theta$ | Discriminative frame | s | Scoring function set |
| $R^l$ | Fuzzy rule $l$ | $N$ | Number of training samples |
| $z_{CA}$ | Inputs of CA | $X_i, V_i, P_i$ | Position, velocity, optimal solution of the $i$-th particle |
| $E_{CA}$ | Universe of discourses of CA | $G$ | Size of particle swarm |
| $A_1^l$ | Intuitionistic fuzzy subsets | $P_g$ | Current global optimal solution |
| $p^l$ | Consequent parameter | $w_{\min}, w_{\max}$ | Minimum, maximum inertia weights |
| $\mu(\bullet), v(\bullet)$ | Membership, non-membership degree | $c_1, c_2$ | Learning parameter |
| $\pi(\bullet)$ | Intuitionistic index | $T$ | Number of iterations |
| $L_{RG}$ | Number of fuzzy rules | $M$ | Number of label vector dimensions |
| $y_{RG}^0$ | Outputs for the model | $f_{\min}, f_{\max}, f_{avg}$ | Minimum, maximum, and average fitness of the particle swarm |

## 2. Preliminaries

In this section, the preliminaries of the Dempster–Shafer evidence theory and Takagi–Sugeno intuitionistic fuzzy rules method are first introduced.

### 2.1. Evidence Theory

Dempster–Shafer evidence theory has flexibility and effectiveness in modeling uncertainties without prior information [19]. A discriminative frame $\Theta$ consisting of all possible propositions is defined as follows:

$$\Theta = \{\theta_1, \theta_2, \cdots, \theta_i, \cdots, \theta_n\} \tag{1}$$

Mass function mapping $m$ from $2^\Theta$ to [0, 1] is defined as BBA, which satisfies the following conditions:

$$m(\varnothing) = 0 \ \ and \ \ \sum_{\theta \subseteq \Theta} m(\theta) = 1 \tag{2}$$

If $m(\theta) > 0$ , then $\theta$ is described as the focal element. Suppose two independent basic belief assignments $m_1, m_2$ construct the form $m_1 \oplus m_2$ according to Dempster's rule of combination, which can be expressed as follows:

$$m(\theta) = \begin{cases} \frac{1}{1-K} \sum\limits_{E \cap F = \theta} m_1(E)m_2(F), & \theta \neq \varnothing \\ 0 & \theta = \varnothing \end{cases} \qquad (3)$$

With

$$K = \sum\limits_{E \cap F \neq \varnothing} m_1(E)m_2(F) \qquad (4)$$

where $E, F \in 2^{\theta}$ and $K$ is the conflict coefficient of $m_1$ and $m_2$. When the evidence is highly conflicting, the evidence fusion processing will lead to counter-intuitionistic results. For a multi-source target recognition system, a degree of conflict of the information is provided by each sensor, so dealing with the conflicts between the evidence is the key to applying various evidence-based theories for accurate target recognition.

The common features of information on aerial targets, such as flight speed, acceleration, flight height and so on, can be detected by a multi-source system. Due to the problem of various forms of signal interference and other factors, a system detecting the target information will contain a lot of uncertainty. Most methods based on decision-level data fusion, such as D-S and Yager, require a high level of data preprocessing and display low interpretability. In order to improve the interpretability of the information fusion and the process of aerial target recognition, the T-S intuitionistic fuzzy model is introduced to establish mapping between the feature space and the target space. The T-S intuitionistic fuzzy model has strong learning ability and robustness, which means it can label historically detected targets with the correct categories, and input their feature information into the T-S intuitionistic fuzzy model for training after intuitionistic fuzzification, then forming a correct mapping relationship. By continuously learning target features, the final trained model can accurately obtain the mapping relationship between the features and the targets.

### 2.2. Takagi–Sugeno Intuitionistic Fuzzy Rules Method

When the number of input variables increases, the number of rules of the T-S model will increase exponentially, resulting in a decrease in training performance. For typical aerial targets, we divide the features of the aerial targets into two or three groups for modeling. Figure 1 illustrates the classification and the process of target recognition.

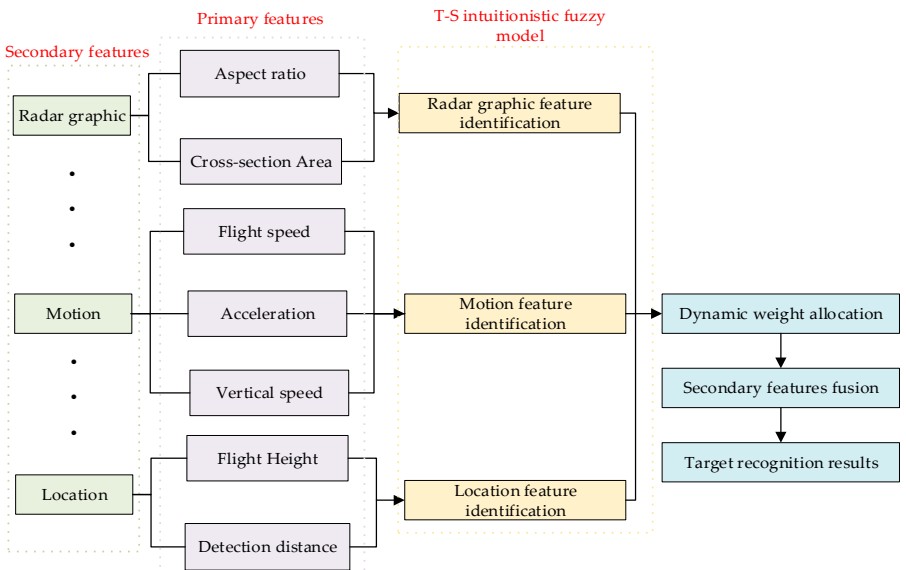

**Figure 1.** Target recognition T-S intuitionistic fuzzy model.

First, the features are divided into primary features and secondary features with the target properties, and each secondary feature contains two or three primary features. Then, the model is trained by the training data to obtain the premise and consequence parameters, and the primary features are fused and judged by the trained MTS-IFRM. Finally, the identity estimation results of the target are fused with secondary features to obtain the final recognition result of the target.

The main difficulty of aerial target recognition lies in the fusion of multiple features. Achieving accurate recognition of targets from imprecise and conflicting feature data is the key. This section will mainly introduce the proposed aerial target recognition algorithm. The MTS-IFRM is designed by taking the radar graphic (RG) as an example, the inputs of the model are the feature values of aspect ratio (AR) and cross-sectional area (CA) after intuitionistic fuzzification, then we define the MTS-IFRM based on a fuzzy set:

$$
\begin{aligned}
&Rule\ R^l:\ If\ z_{CA}\ is\ A_1^l,\ and\ z_{AR}\ is A_2^l,\ then: \\
&f_{RG}^l(z_{RG}) = p_{RG0}^l + p_{RG}^l S(z_{CA}) + p_{RG}^l S(z_{AR}), l = 1, 2, \ldots, L_{RG}
\end{aligned} \tag{5}
$$

where the part after "*if*" denotes the premise and the part after "*then*" denotes the consequence of the rule. $z_{CA} = \{\langle CA, \mu(CA), v(CA)\rangle | CA \in E_{CA}\}$ and $z_{AR} = \{\langle AR, \mu(AR), v(AR)\rangle | AR \in E_{AR}\}$ denote the inputs of the CA and AR after intuitionistic fuzzification, respectively. $\mu(\bullet)$ and $v(\bullet)$ are the degrees of membership and the non-membership, respectively, which represent the intuitionistic fuzzy number. Then, $0 \le \mu(\bullet) + v(\bullet) \le 1$. $\pi(\bullet) = 1 - \mu(\bullet) - v(\bullet)$ denotes the intuitionistic index of the intuitionistic fuzzy number. The specific process can be referenced in [36]. $E_{CA}$ and $E_{AR}$ denote the universe of discourses of the CA and AR, respectively. $A_1^l$ and $A_2^l$ denote the intuitionistic fuzzy subsets corresponding to the inputs $z_{CA}$ and $z_{AR}$ of rule $l$, respectively. The input vector $z_{RG} = [z_{CA}, z_{AR}]$ denotes the premise variable of the model. $p_{RG}^l = \left[p_{RG0}^l, p_{RG1}^l, p_{RG2}^l\right]$ denotes the consequence part. $S(\bullet)$ denotes the scoring function with the abilities of sequencing and decision-making, which converts an intuitionistic fuzzy set into a definite numerical value [37]. $L_{RG}$ denotes the RG number of fuzzy rules. Therefore, the weighted average $y_{RG}^0$ of the final outputs for each rule $f_{RG}^l(z_{RG})$ are obtained by:

$$
y_{RG}^0 = \sum_{l=1}^{L_{RG}} \frac{\mu^l(z_{RG}) f_{RG}^l(z_{RG})}{\sum_{\widetilde{l}=1}^{L_1} \mu^{\widetilde{l}}(z_{RG})} = \sum_{l=1}^{L_{RG}} \widetilde{\mu}^l(z_{RG}) \cdot f_{RG}^l(z_{RG}) \tag{6}
$$

where $\mu^l(z_{RG})$ denotes the fuzzy membership degree of fuzzy rule $l$ to input $z_{RG}$. The normalization method is defined as:

$$
\widetilde{\mu}^l(z) = \frac{\mu^l(z_{RG})}{\sum_{\widetilde{l}=1}^{L_{RG}} \mu^{\widetilde{l}}(z_{RG})} \tag{7}
$$

where

$$
\mu^l(z_{RG}) = \mu_{A_1^l}(z_{CA}) \cdot \mu_{A_2^l}(z_{AR}) \tag{8}
$$

$$
\widetilde{\mu}^l(z_{RG}) = \frac{\mu^l(z_{RG})}{\sum_{\widetilde{l}=1}^{L_{RG}} \mu^{\widetilde{l}}(z_{RG})} \tag{9}
$$

$$
\mu_{A_1^l}(z_{CA}) = \lambda_1 \mu_{A_1^l}(z_{CA}) + \lambda_2 v_{A_1^l}(z_{CA}) + \lambda_3 \pi_{A_1^l}(z_{CA}) \tag{10}
$$

$$
\mu_{A_1^l}(z_{AR}) = \lambda_1 \mu_{A_1^l}(z_{AR}) + \lambda_2 v_{A_1^l}(z_{AR}) + \lambda_3 \pi_{A_1^l}(z_{AR}) \tag{11}
$$

Here, $\mu_{A_i^l}(\bullet)$, $v_{A_i^l}(\bullet)$ and $\pi_{A_i^l}(\bullet)$ are calculated by the premise parameter identification. $\mu_{A_i^l}(\bullet)$ can be expressed by using a suitable index $\lambda$ (generally setting $\lambda_1 = 1$, $\lambda_2 = 0$, and $\lambda_3 = 0.5$).

Similarly, the MTS-IFRM based on the feature of motion (M) and location (L) can be established. The output results of the corresponding model are defined as follows:

$$y_{MF}^0 = \sum_{l=1}^{L_M} \frac{\mu^l(z_M) f_M^l(z_M)}{\sum_{\tilde{l}=1}^{L_M} \mu^{\tilde{l}}(z_M)} = \sum_{l=1}^{L_M} \tilde{\mu}^l(z_M) \cdot f_M^l(z_M) \tag{12}$$

$$y_L^0 = \sum_{l=1}^{L_L} \frac{\mu^l(z_3) f_L^l(z_L)}{\sum_{\tilde{l}=1}^{L_3} \mu^{\tilde{k}}(z_L)} = \sum_{l=1}^{L_L} \tilde{\mu}^l(z_L) \cdot f_{LF}^l(z_L) \tag{13}$$

where $z_M = [z_{FS}, z_A, z_{VS}]$, $z_L = [z_{FH}, z_{DD}]$, $z_{FS}, z_A, z_{VS}, z_{FH}$ and $z_{DD}$ denote flight speed, acceleration, vertical speed, flight height, and detection distance features after intuitionistic fuzzification, respectively.

## 3. Aerial Target Recognition Methods Based on the MTS-IFRM

According to the above analysis, parameter identification is a central role of a T-S rule-based system, which evaluates the quality of the rule modeling. Therefore, the related work of the MTS-IFRM contains the structure identification of consequent parameters based on the ridge regression method, the identification of the premise part with a novel intuitionistic fuzzy C-means (IFCM) clustering model, and the adaptive weight algorithm.

### 3.1. Construction of MTS-IFRM

In this section, we take the training of the RG consequence parameters as an example. First, according to Equations (5) and (6), let:

$$s_e = \left(1, s^T\right)^T \tag{14}$$

where $s = [S(z_{CS}), S(z_{AR})]$ denotes the scoring function set of the input z. So that:

$$\tilde{s}_{RG}^l = \tilde{\mu}^l(z_{RG}) s_e \tag{15}$$

$$s_{g,RG} = \left(\left(\tilde{s}_{RG}^1\right)^T, \left(\tilde{s}_{RG}^2\right)^T, \ldots, \left(\tilde{s}_{RG}^{L_{RG}}\right)^T\right)^T \tag{16}$$

$$p_{RG}^l = \left(p_{RG}^l, p_{RG}^l, p_{RG}^l\right)^T \tag{17}$$

$$p_{g,RG} = \left(\left(p_{RG}^1\right)^T, \left(p_{RG}^2\right)^T, \ldots, \left(p_{RG}^{L_{RG}}\right)^T\right)^T \tag{18}$$

where $\tilde{\mu}^l(z_{RG})$ is acquired in Equation (7). Next, the output of the model is denoted as:

$$y_{RG}^0 = \left(p_{g,RG}\right)^T s_{g,RG} \tag{19}$$

In Equation (19), we obtain the RG output of the MTS-IFRM. To solve the target recognition problem, each secondary feature needs to have the corresponding output. Therefore, the MTS-IFRM is constructed. The ridge regression method, a modified analysis of the least-squares estimation, can deal with multicollinearity by operating the unbiased estimator. To obtain a more reliable estimate of the consequent parameter, ridge regression analysis is constructed to train the model:

$$\min_{p_{g,m,RG}} J(p_{g,m,RG}) = \frac{1}{2} \sum_{m=1}^M \sum_{n=1}^N \left(\left(p_{g,m,RG}\right)^T s_{g,n,RG} - \tilde{y}_{n,m}\right)^2 + \gamma_1 \sum_{m=1}^M \sum_{n=1}^N \left(p_{g,m,RG}\right)^T p_{g,m,RG} \tag{20}$$

Equation (20) contains the minimization of empirical risk and structure risk. Where $p_{g,m,RG}$ denotes the consequent parameter of the *m*-th aerial target, $N$ denotes the number of training samples, $\tilde{y}_{n,m}$ denotes the *M*-dimensional label vector of the *n*-th training sample,

$\gamma_1$ represents the regularization parameter. To adjust the consequent parameter $p_{g,m,RG}$, the final optimization result is calculated by the first-order necessary condition:

$$\frac{\partial J\left(p_{g,m,RG}\right)}{\partial p_{g,m,RG}} = \sum_{m=1}^{M}\sum_{n=1}^{N}\left(s_{g,n,RG}(s_{g,n,RG})^T + \gamma_1 I_{l \times l}\right) \cdot p_{g,m,RG} - \sum_{m=1}^{M}\sum_{n=1}^{N}\left(s_{g,n,RG}\widetilde{y}_{n,m}\right) = 0 \quad (21)$$

In Equation (21), $p_{g,m,RG}$ is as follows:

$$p_{g,m,RG} = \sum_{n=1}^{N}\left(\gamma_1 I_{l \times l} + s_{g,n,RG}(s_{g,n,RG})^T\right)^{-1}\sum_{m=1}^{M}\sum_{n=1}^{N}\left(s_{g,m,RG}\widetilde{y}_{n,m}\right) \quad (22)$$

Therefore, a new MTS-IFRM of RF for aerial target recognition can be expressed as follows according to Equations (5) and (22):

$$\begin{gathered}\text{Rule } R^{l'}: \text{ If } z'_{CS} \text{ is } A_1^{l'}, \text{ and } z'_{AR} \text{ is} A_2^{l'}, \text{ then}: \\ f_{RG}^{l'}(z'_{RG}) = p_{g,j,RG0}^{l'} + p_{g,j,RG1}^{l'}S(z'_{CS}) + p_{g,j,RG2}^{l'}S(z'_{AR}), l' = 1, 2, \ldots, L'_{RG}\end{gathered} \quad (23)$$

where $z'_{CS}$ and $z'_{AR}$ are the CS features and AR features after intuitionistic fuzzification, respectively. $p_{g,m,RGi}^{l'}$ denotes the consequent parameter corresponding to rule $l'$ of model $m$, here, $i = 0, 1, 2$ and $L'_{RG}$ denotes the number of rules. Similarly, the corresponding rules of the MTS-IFRM for the motion feature (MF) and location feature (LF) can be established in the same construction procedures.

### 3.2. Premise Identification

IFCM and the FCM clustering are very sensitive to the initial clustering center position and are prone to converging to the local optimal solution in a noisy environment. Moreover, the variation factors of dynamic evolution theory are introduced into the PSO algorithm to improve the clustering optimization model [38].

Suppose the position of the $i$-th particle is $X_i = (x_{i,1}, x_{i,2}, \ldots, x_{i,d})$, the velocity is $V_i = (v_{i,1}, v_{i,2}, \ldots, v_{i,d})$ and $P_i = (p_{i,1}, p_{i,2}, \ldots, p_{i,d})$ is the optimal solution in $d$-dimensional space, where $i = 1, 2, \cdots, G$, $G$ is the size of the particle swarm, then the velocity and position updated in the $j$-th dimension at an iteration are:

$$v_{i,j}(t+1) = \omega v_{i,j}(t) + c_1 r_1(p_{i,j} - x_{i,j}(t)) + c_2 r_2(p_{g,j} - x_{i,j}(t)) \quad (24)$$

$$x_{i,j}(t+1) = x_{i,j}(t) + v_{i,j}(t+1), \quad j = 1, 2, \cdots, d \quad (25)$$

where $P_g = (p_{g,1}, p_{g,2}, \ldots, p_{g,d})$ denotes the current global optimal solution, $w$ denotes the inertia weight. $r_1$ and $r_2$ are random numbers in the interval [0, 1], respectively. $c_1$ and $c_2$ denote the learning parameter of the DPSO, respectively, and are defined as follows:

$$c_1 = 2.5 - 2 \times \frac{t}{T} \quad (26)$$

$$c_2 = 2.5 + 2 \times \frac{t}{T} \quad (27)$$

where $t$ denotes the number of iterations in this round, and $T$ denotes the maximum number of iterations. $c_1$ and $c_2$ change dynamically to meet the changing rule with the increase in the number of iterations. Therefore, the algorithm can adaptively expand the local search range in the early stage of iteration and accelerate the global convergence speed in the late iteration. This learning mechanism is used to accelerate the overall convergence.

In the iterative process, the inertia weight can affect the search range of the current round according to the speed of the previous round. At the end of each round of iterations, the fitness function of the selected particle swarms should be obtained. Moreover, the inertia weights can be dynamically adjusted based on the results of the fitness values, which will make the selected particle swarms in this round of iterations have a more balanced

position. The nonlinear adaptive inertia weight strategy is used to calculate the inertia weight, and the method is as follows:

$$
w = \begin{cases} w_{\min} + \frac{(w_{\max} - w_{\min}) \times (f_i - f_{\min})}{f_{avg} - f_{\min}}, f_i \leq f_{avg} \\ w_{\max}, f_i > f_{avg} \end{cases} \tag{28}
$$

where $w_{\max}$ and $w_{\min}$ are the maximum and minimum inertia weights set, respectively, and $f_{\min}$ and $f_{\max}$ represent the minimum and maximum fitness values of the particle swarm in this round, respectively. $f_{avg}$ represents the average fitness of a particle swarm. At this point, the speed of the particle swarm mainly refers to the speed of the previous round to increase the activity of the particle swarm. Conversely, the speed of the particle swarm at this time mainly refers to the local optimal position and the global optimal position to accelerate the particle swarm to move closer to the dominant space.

Suppose $Z = \{\mathbf{z}_1, \mathbf{z}_2, \ldots, \mathbf{z}_N\}$ is the dataset, where $\mathbf{z}_n = [z_1, z_2, \ldots, z_d]^T$ and $z_i = \{\langle x_i, \mu(x_i), v(x_i)\rangle | x_i \in E\}$, $1 \leq i \leq d$. $N$ is the number of data items. $m$ is the number of clusters. Here, $V = \{v_1, v_2, \ldots, v_m\}$, $v_m \in R^d$, is a set of $M$ clustering centers where $M \geq 2$. Each clustering center vector can be expressed as $v_m = [c_1^m, c_2^m, \ldots, c_d^m]$, where $c_i^m = \{\langle c_i^m, \mu_{v_m}(c_i^m), v_{v_m}(c_i^m)\rangle | 1 \leq i \leq d, 1 \leq m \leq M\}$. The objective function is given below:

$$
\begin{aligned}
J_m(U, V) &= \sum_{n=1}^{N} \sum_{m=1}^{M} \mu_{nm}{}^{c_0} d_{nm}{}^2(z_n, v_m) \\
\mu_{nm} &\in [0, 1], \ 1 \leq m \leq M, \ 1 \leq n \leq N \\
\sum_{m=1}^{M} \mu_{nm} &= 1, \ \forall n, m
\end{aligned} \tag{29}
$$

where $\mu_{nm}$ is the membership degree of the sample data in the $m$-th class. $U = [\mu_{nm}]_{N \times M}$ denotes the fuzzy membership matrix of $X.c_0 \in [1, +\infty)$ denotes the fuzzification index. $d_{nm}{}^2(z_n, v_m)$ denotes the ordinary Euclidean distance between the measurement point $z_n$ and the clustering center $v_m$, which is defined as:

$$
d_{nm}{}^2(z_n, v_m) = \frac{1}{2} \sum_{i=1}^{d} p_i \left\{ [\mu_{z_n}(x_i) - \mu_{v_m}(c_i^m)]^2 + [v_{z_n}(x_i) - v_{v_m}(c_i^m)]^2 + [\pi_{z_n}(x_i) - \pi_{v_m}(c_i^m)]^2 \right\} \tag{30}
$$

where $p_i = (1/p, 1/p, \ldots, 1/p)$, $\mu_{z_n}(x_i)$, $v_{z_n}(x_i)$ and $\pi_{z_n}(x_i)$ are the fuzzy membership degree, non-membership degree, and intuitionistic index of input data $z_n$, respectively. $\mu_{v_k}(c_i^m)$, $v_{v_k}(c_i^m)$ and $\pi_{v_k}(c_i^m)$ are the fuzzy membership degree, non-membership degree, and intuitionistic index of clustering center $v_m$, respectively.

Therefore, to obtain the optimal objective function by DPSO, it can be considered that the smaller the result of the objective function $J_m(U, V)$, the better the fitness of the particles, so the particle fitness can be expressed by the following:

$$
f(x_i) = \frac{\lambda}{J(U, V)} = \frac{\lambda}{\sum\limits_{n=1}^{N} \sum\limits_{m=1}^{M} \mu_{nm}{}^2 d_{nm}{}^2(z_n, v_{i,m})} \tag{31}
$$

where $v_{i,m}$ denotes the intuitionistic fuzzy number of the $m$-th dimension of particle $x_i$ and also denotes $m$-th clustering center. $\lambda$ is a constant, which can be manually adjusted according to the specific situation. The main steps for DPSO-IFCM are summarized in Figure 2.

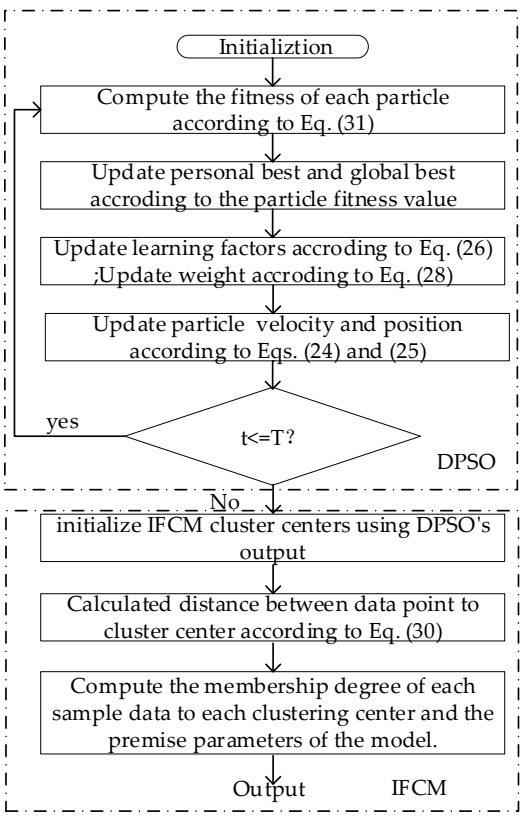

**Figure 2.** DPSO-IFCM algorithm processes.

In Figure 2, it is shown that the proposed DPSO-IFCM clustering algorithm includes the following steps:

1. Initialization: Initialize $G$ particles to form $G$ first-generation particles, where each particle randomly generates $M$ clustering centers. The fitness value is calculated by Equation (31) and determines the current optimal position of each particle $i$ by the fitness value, and the position of the current particle swarm with the highest fitness is $p_g$;

2. Compute the velocity and position of each particle in the new particle swarm using Equations (24) and (25);

3. Compute the fitness value of each particle in the new particle swarm using Equation (31) and compare it with the previous generation. For the same individual, if the individual fitness in the new population is larger than the corresponding individual in the previous generation, replace the individual of the previous generation and this becomes the optimal position of particle $i$, otherwise, it remains unchanged;

4. Compare the fitness value of the optimal individual of the new particle swarm with the optimal individual of the previous generation, if the fitness is greater than the previous generation, update the optimal position of the population to the optimal position of the new particle swarm, otherwise, it remains unchanged, then $t = t + 1$.

5. Repeat Steps 2–4 until a criterion is met that is usually of a sufficiently good fitness or a maximum number of iterations;

6. Obtain the individual position with the highest fitness value as the initial clustering center of the IFCM algorithm;

7. Compute the membership degree $\mu_{nm}$ of each sample dataset to each clustering center and the premise parameters $\mu_{A_i^m}(x_i), v_{A_i^m}(x_i), \pi_{A_i^m}(x_i)$ of the model. A detailed method can be found in Ref. [36].

Finally, we input the intuitionistic fuzzy features into the trained MTS-IFRM. The output of the *j*-th model is:

$$\widetilde{y}_j = \sum_{l'=1}^{L'} \frac{\mu^{l'}(\mathbf{z})f^{l'}(\mathbf{z})}{\sum_{i=1}^{L'}\mu^i(\mathbf{z})} = \sum_{l'=1}^{L'} \widetilde{\mu}^{l'}(\mathbf{z}) \cdot f^{l'}(\mathbf{z}) \tag{32}$$

### 3.3. Adaptive Weight Algorithm

From Equation (32), we know that every target has a corresponding MTS-IFRM, then each model is trained and obtains the corresponding label vector output. If the features of the input data are more similar to a certain class, then the value of the corresponding class in the label vector output will be closer to one, otherwise, the value will be closer to zero. When the values of more than one class are relatively close, the class cannot be well distinguished from the input features; that is, the degree of discrimination is not obvious. At this point, we can focus on other models to realize the classification and recognition of the target; that is, reduce the weight corresponding to the model with a low degree of discrimination, and increase the weight corresponding to the model with a high degree of discrimination. First, the initial weight of each model is $1/h$, $h$ denotes the number of secondary features, the weight distribution is also related to the following two points:

1.  For a certain secondary feature, in the output result of the corresponding model, if all the values in the output vector are less than 0.5, the possibility of the feature belonging to the target being classified is too low. Therefore, the secondary feature should be reduced according to the impact of the secondary features on the classification results, the weight corresponding to the secondary features is reduced and assigned to other features. Suppose that the maximum value of the label vector output by the model is $x_{\max}$, the weight of the corresponding model can be expressed as:

$$S_1(x_{\max}) = \frac{1}{1 + e^{h_1 \cdot (h_2 - x_{\max})}} \tag{33}$$

The final output matrix can be obtained:

$$f_1(x_{\max}) = \frac{1}{h} \cdot \frac{1}{1 + e^{h_1 \cdot (h_2 - x_{\max})}} \tag{34}$$

where $h_1$ and $h_2$ are two constants to control the speed of weight change. Figure 3 shows the weight change under $h_1 = 20, h_2 = 0.25$.

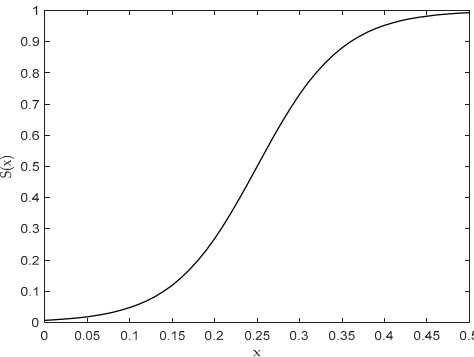

**Figure 3.** Weight adjustment in case 1.

In Figure 3, when $x_{\max}$ is less than 0.5, the weight of the corresponding model will gradually decrease. When $x_{\max} = 0.25$, the weight of the corresponding model will decrease rapidly. When the weight is below 0.1, the corresponding model weight is close to 0 and the larger weight will be allocated to the model that can be better identified, which can obtain a higher recognition accuracy.

2. For a certain secondary feature to the corresponding T-S IFM output, if the maximum value in the label vector is greater than 0.5, and the difference between the maximum value and the second large value is less than 0.3, then the classification ability of the secondary features for all of the targets to be classified is weak. However, because the maximum value in the label vector is greater than 0.5, the feature has a certain classification ability for a certain type or several types of targets, but it cannot determine which type the input feature data belongs to. Therefore, the corresponding weight can be appropriately reduced and assigned to other features.

Suppose that the difference between the maximum value and the sub-maximum value in the label vector output by the model is $x_{dif} = x_{first} - x_{second}$, $0 \leq x_{dif} \leq 0.3$. Finally, in case 2, Figure 4 shows the weight adjustment under $h_1 = 20, h_2 = 0$. Different from case 1, case 2 cannot clearly distinguish which category the target belongs to, because there is a value in the label vector, only the weight is appropriately reduced. From Figure 4, the weight is reduced to at most half of the original.

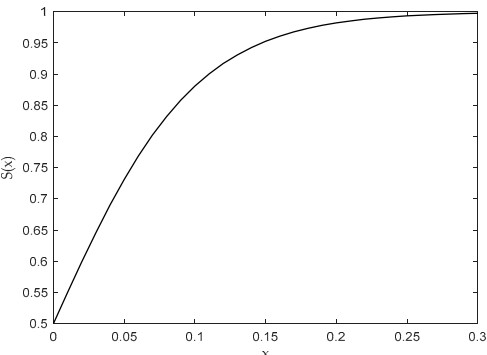

**Figure 4.** Weight adjustment in case 2.

According to the above two points of analysis, the final weight allocation method of each model is designed, and the process is as follows:

To assign the reduced weight portion of the model of cases 1 and 2 equally to the other models, first, the number of secondary features that do not satisfy the above two cases can be expressed as:

$$num = \begin{cases} num, & x_{\max} \leq 0.5 \ or \ x_{dif} \leq 0.3 \\ num + 1, & 0.5 < x_{\max} \leq 1 \ and \ 0.3 < x_{dif} \leq 1 \end{cases} \tag{35}$$

Equation (35) denotes the number of models with obvious classification effects. Then, the final weight adjustment of each model can be expressed as:

$$W_i = \begin{cases} f_{\max}(x_{\max}), & x_{\max} \leq 0.5 \\ f_2(x_{dif}), & 0.5 < x_{\max} \leq 1 \ and \ x_{dif} \leq 0.3 \\ \frac{1}{n} + \frac{1}{num} \sum\limits_{j=1, j \neq i}^{n} (1 - f(x_j)), & 0.5 < x_{\max} \leq 1 \ and \ 0.3 < x_{dif} \leq 1 \end{cases} \tag{36}$$

where $W_i$ denotes the final weight of the *i-th* model. $f(x_j)$ denotes the weight of the corresponding model when case 1 or case 2 occurs. Therefore, the final fusion results are calculated as follows:

$$y^0 = W_{RF} y_{RF}^0 + W_{MF} y_{MF}^0 + W_{LF} y_{LF}^0 \tag{37}$$

*3.4. Computational Complexity Analysis*

In the proposed MTS-IFRM, the main program includes the implementation of the DPSO-IFCM algorithm and the structure identification of consequence parameters based on the ridge regression method. The total computational complexity of the ridge regression

is calculated as N($L \cdot N \cdot M^2$), where $L$ is the number of intuitionistic T-S fuzzy rules, $N$ is the number of samples, and $M$ is the number of label vector dimensions. In the DPSO-IFCM algorithm, the total computational complexity of the main loop of DPSO is N($G \cdot T \cdot d$), where $G$ is the size of the particle swarm, $T$ is the maximum number of iterations, $d$ is the dimension of the solution space, and the calculation time of the IFCM is mainly used for the fuzzy membership $\mu_{nm}$ and the computational complexity is N($L \cdot N \cdot M$). In summary, the computational cost of the proposed algorithm is determined by $L$, $N$, $M$, $G$, $T$, and $d$.

## 4. Simulation Results and Analysis

To evaluate the performance of the MTS-IFRM approach to the problem of recognizing aerial targets in a complex environment, two examples were used to show the recognition performance of MTS-IFRM compared to that of the standard forms of the D-S [19], Yager [29], Murphy [30], multi-sensor data fusion algorithm (MSDF) [32], Kaur [8], and Hu [9] in a complicated environment. Table 2 presents the feature ranges of five typical aerial targets (bomber (Br), fighter (Fr), helicopter (Hr), air-to-ground missile (AGM), and tactical ballistic missile (TBM)).

**Table 2.** Feature ranges of five aerial targets.

|  | **Br** | **Fr** | **Hr** | **AGM** | **TBM** |
|---|---|---|---|---|---|
| Flight height (km) | 25–35 | 7–13 | 1.6–2.5 | 3.8–5.2 | 55–80 |
| Detection distance (km) | 350–450 | 250–350 | 130–180 | 100–140 | 130–180 |
| Flight speed (m/s) | 300–500 | 500–700 | 70–130 | 1000–1500 | 1700–2300 |
| Acceleration (m/s$^2$) | 0–20 | 0–50 | 0–30 | 150–250 | 200–400 |
| Vertical speed (m/s) | 0–50 | 0–300 | 0–50 | 800–1200 | 1600–2300 |
| Cross-section area (m$^2$) | 0.25–0.35 | 0.17–0.23 | 0.08–0.12 | 0.05–0.08 | 0.06–0.11 |
| Aspect ratio | 1.2–2.0 | 2.6–3.6 | 3.2–4.8 | 6.7–9.3 | 8.5–11.5 |

Table 2 shows the complete discernment frame is $\Theta = \{Br, Fr, HG, AGM, TBM\}$, and the target recognition feature set is $\mathbf{E} = \{E_A, E_B, E_C, E_D, E_E, E_F, E_G\}$, which represents the credibility of the evidence of the flight altitude (FH), detection distance (DD), flight speed (FS), acceleration (A), vertical speed (VS), cross-section area (CA), and aspect ratio (AR), respectively. The training data is generated within the scope of feature ranges, the experiment uses 125 sets of target feature data within the appropriate range as the training phase with the rules of nine sets. Table 3 presents seven training datasets from the training datasets.

**Table 3.** The feature data of aerial targets.

| Serial Number | FH (km) | DD (km) | FS (m/s) | A (m/s$^2$) | VS (m/s) | CA (m$^2$) | AR | Target |
|---|---|---|---|---|---|---|---|---|
| 1 | 58.6 | 135.6 | 1845.5 | 210.4 | 1685.6 | 0.08 | 9.5 | TBM |
| 2 | 4.2 | 125.8 | 1250.7 | 211.9 | 952.2 | 0.06 | 8.4 | AGMM |
| 3 | 8.3 | 344 | 612.3 | 42.6 | 258.4 | 0.22 | 2.7 | Fr |
| 4 | 4.5 | 132.8 | 1252.1 | 158.7 | 958.8 | 0.06 | 6.9 | AGM |
| 5 | 31.6 | 377.4 | 315.3 | 14.6 | 25.3 | 0.31 | 1.6 | Br |
| 6 | 1.7 | 136.6 | 88.6 | 11.3 | 29.3 | 0.09 | 3.8 | Hr |
| 7 | 56.6 | 179.1 | 2200.6 | 365.6 | 1936.7 | 0.10 | 11.5 | TBM |

Table 3 shows that the collected 13–14 d historical feature datasets with results are obtained as the training datasets and the testing target is recognized according to the trained MTS-IFRM, then the feature datasets and model parameters of the target are updated with the recognition result.

The fuzzy membership function is very important for the initial recognition process because of the uncertainty in the feature data. By analyzing the features of aerial targets, the Gaussian membership function is used to recognize the target in Equation (38) and

Table 4 presents $\delta$ and $x$ of five typical aerial targets with difference features, showing the fuzzy membership functions corresponding to the detection distance.

$$\mu(x_i) = \exp\left(-\frac{\|x - x_i\|}{\delta}\right) \tag{38}$$

**Table 4.** Five typical aerial targets with different features.

|  | **Br** | **Fr** | **Hr** | **AGM** | **TBM** |
|---|---|---|---|---|---|
| FH (km) | (30,7.5) | (10,4.5) | (2,1) | (4.5,1) | (65,15) |
| DD (km) | (400,80) | (300,80) | (200,60) | (120,45) | (150,60) |
| FS (m/s) | (400,150) | (600,150) | (100,50) | (1200,500) | (2000, 500) |
| A(m/s$^2$) | (10,10) | (25,25) | (15,15) | (200,60) | (300,100) |
| VS(m/s) | (25,25) | (150,150) | (25,25) | (1000,300) | (1950,600) |
| CA (m$^2$) | (0.3,0.08) | (0.2,0.06) | (0.1,0.03) | (0.06,0.02) | (0.08,0.03) |
| AR | (1.5,0.5) | (3,0.6) | (4,0.8) | (8,1.3) | (10,1.5) |

Table 4 shows the appropriate membership function $\mu(x_i)$ can be designed by adjusting $\delta$ and $x$ with the different features of the targets $x_i$ by analyzing the various feature attributes of each target in Table 2. Then, take the feature of detection distance as an example. Figure 5 presents the fuzzy membership functions corresponding to the detection distance.

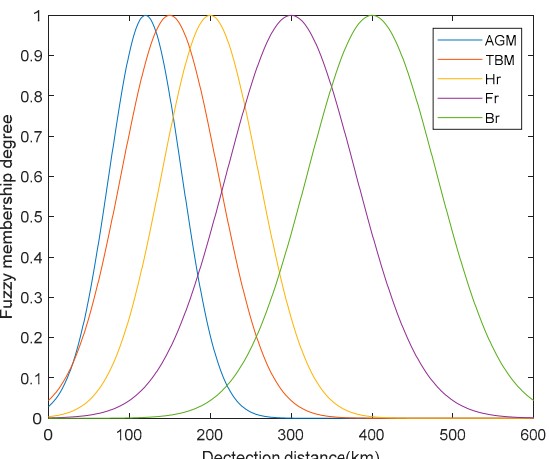

**Figure 5.** Fuzzy membership functions of detection distance.

From Figure 5, the fuzzy membership degree of each target will be different with different values of primary features. When the detection distance is 450 km, the fuzzy membership degree belonging to target Br is the highest, which is 0.8226, and the fuzzy membership degree belonging to the target AGM is the lowest, approaching zero. When the target features obtained by the radar system are inaccurate and uncertain, the features are calculated by the membership function, thus effectively recognizing the target initially. Figure 6 shows the target recognition framework based on fuzzy membership degree and evidence theory.

In Figure 6, the supporting information of the target obtained by the fuzzy membership function may not be consistent. We use the recognition result of the target obtained by the fuzzy membership function as the confidence degree, and evidence theory is used to fuse the confidence degree and obtain a target recognition result.

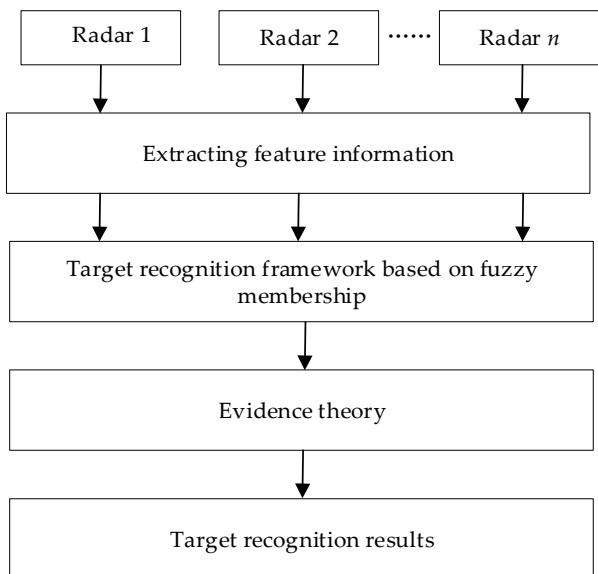

**Figure 6.** The target recognition framework based on fuzzy membership and evidence theory.

*4.1. Example 1: The Data Does Not Contain Fault Features*

In this example, data without fault features is employed to show the performance of the methods, that is, all target features support a certain target. Suppose the radar detects a suspicious target, the target features are: $A = 23$ km, $B = 450$ km, $C = 350$ m/s, $D = 10$ m/s$^2$, $E = 40$ m/s, $F = 0.31$ m$^2$, and $G = 1.0$. Table 5 presents the BPA example of multi-source information fusion.

**Table 5.** The BPA example of the multi-source information fusion.

| Evidence | Br | Fr | Hr | AGM | TBM | X |
|----------|-----|-----|-----|-----|-----|---|
| $E_A$ | 0.4185 | $2.37 \times 10^{-4}$ | 0 | 0 | $3.93 \times 10^{-4}$ | 0.5809 |
| $E_B$ | 0.6766 | 0.0297 | $2.88 \times 10^{-4}$ | 0 | 0 | 0.2936 |
| $E_C$ | 0.8837 | 0.0297 | 0 | 0.0549 | $1.84 \times 10^{-5}$ | 0 |
| $E_D$ | 0.3857 | 0.0614 | 0.3451 | $1.70 \times 10^{-5}$ | $8.59 \times 10^{-5}$ | 0 |
| $E_E$ | 0.3525 | 0.2691 | 0.3525 | $1.80 \times 10^{-5}$ | $2.01 \times 10^{-5}$ | 0 |
| $E_F$ | 0.9660 | 0.0340 | 0 | 0 | 0 | 0 |
| $E_G$ | 0.3679 | $1.49 \times 10^{-5}$ | $7.81 \times 10^{-5}$ | 0 | 0 | 0.6321 |

Table 5 shows the corresponding BPA functions and *X* denotes the unknown term. The features are expressed with fuzzy membership for the unknown targets detected by radar, all the features of the unknown target have high credibility for the target Br, and no feature opposes the Br. Tables 6–9 show the recognition results of the target with different numbers of evidence in an error-free environment.

**Table 6.** Comparison of algorithms with $E_A$ and $E_B$ in an error-free environment.

| Method | m(Br) | m(Fr) | m(Hr) | m(AGM) | m(TBM) | m(X) | Target |
|--------|-------|-------|-------|--------|--------|------|--------|
| D-S | 0.8095 | 0.0176 | 0 | 0 | 0 | 0.1728 | Br |
| Yager | 0.2832 | 0 | 0 | 0 | 0 | 0.7168 | X |
| Murphy | 0.7905 | 0.0705 | 0.0715 | 0.0047 | 0 | 0.0627 | Br |
| MSDF | 0.7946 | 0.0692 | 0.0694 | 0.0047 | 0 | 0.0621 | Br |
| Kaur | 0.8056 | 0.0862 | 0.0891 | 0.0056 | 0 | 0.0135 | Br |
| Hu | 0.8134 | 0.0923 | 0.0451 | 0.0026 | 0 | 0.0466 | Br |
| MTS-IFRM | 0.9894 | 0.0064 | 0.0042 | 0 | 0 | 0 | Br |

**Table 7.** Comparison of algorithms with $E_C$, $E_D$ and $E_E$ in an error-prone environment.

| Method | m(Br) | m(Fr) | m(Hr) | m(AGM) | m(TBM) | m(X) | Target |
|--------|-------|-------|-------|--------|--------|------|--------|
| D-S | 0.9610 | 0.0390 | 0 | 0 | 0 | 0 | Br |
| Yager | 0.1201 | 0.0049 | 0 | 0 | 0 | 0.8750 | X |
| Murphy | 0.9020 | 0.0385 | 0.0391 | 0.0021 | 0 | 0.0183 | Br |
| MSDF | 0.9050 | 0.0374 | 0.0375 | 0.0021 | 0 | 0.0180 | Br |
| Kaur | 0.9156 | 0.0395 | 0.0357 | 0.0035 | 0 | 0.0057 | Br |
| Hu | 0.9265 | 0.0402 | 0.0315 | 0.0018 | 0 | 0 | Br |
| MTS-IFRM | 0.9342 | 0.0187 | 0.0472 | 0 | 0 | 0 | Br |

**Table 8.** Comparison of algorithms with $E_F$ and $E_G$ in an error-free environment.

| Method | m(Br) | m(Fr) | m(Hr) | m(AGM) | m(TBM) | m(X) | Target |
|--------|-------|-------|-------|--------|--------|------|--------|
| D-S | 0.9782 | 0.0218 | 0 | 0 | 0 | 0 | Br |
| Yager | 0.3554 | 0 | 0 | 0 | 0 | 0.6446 | X |
| Murphy | 0.7905 | 0.0705 | 0.0715 | 0.0047 | 0 | 0.0627 | Br |
| MSDF | 0.7946 | 0.0692 | 0.0694 | 0.0047 | 0 | 0.0621 | Br |
| Kaur | 0.8165 | 0.0712 | 0.0718 | 0.0056 | 0 | 0.0349 | Br |
| Hu | 0.8564 | 0.0522 | 0.0559 | 0.0062 | 0 | 0.0293 | Br |
| MTS-IFRM | 0.9790 | 0.0210 | 0 | 0 | 0 | 0 | Br |

**Table 9.** Comparison of algorithms with **E** in an error-free environment.

| Method | m(Br) | m(Fr) | m(Hr) | m(AGM) | m(TBM) | m(X) | Target |
|--------|-------|-------|-------|--------|--------|------|--------|
| D-S | 0.9998 | $1.75 \times 10^{-4}$ | 0 | 0 | 0 | 0 | Br |
| Yager | 0.0121 | 0 | 0 | 0 | 0 | 0.9879 | X |
| Murphy | 0.9970 | 0.0014 | 0.0014 | 0 | 0 | 0 | Br |
| MSDF | 0.9973 | 0.0013 | 0.0013 | 0 | 0 | 0 | Br |
| Kaur | 0.9981 | 0.0010 | $9 \times 10^{-4}$ | 0 | 0 | 0 | Br |
| Hu | 0.9985 | 0.0008 | 0.0011 | 0 | 0 | 0 | Br |
| MTS-IFRM | 0.9813 | 0.0015 | 0.0172 | 0 | 0 | 0 | Br |

From Tables 6–9 when the quantity of evidence increases, the recognition accuracy of the other six methods steadily improves except for Yager. The reason is that Yager assigns all the conflicts between evidence to X, which leads to cumulative conflicts between pieces of evidence in the synthetic evidence, and the value of X will increase as the quantity of fusing conflicting evidence increases. When the quantity of evidence is small, the MTS-IFRM maintains better target recognition performance and faster convergence because it can deal with the uncertainty well. Regardless of whether fewer features or more features are available, the MTS-IFRM has higher accuracy when recognizing the targets.

### 4.2. Example 2: The Data Contains Fault Features

The dataset simulated in this paper contains one or more fault features obtained by the equipment, so that the multiple features do not all support a certain target. Suppose the radar detects a suspicious target, the obtained target features are: $A = 23$ km, $B = 450$ km, $C = 350$ m/s, $D = 10$ m/s$^2$, $E = 40$ m/s, $F = 0.31$ m$^2$, and $G = 4.1$. Except for the target aspect ratio, other features are the same as in example 1. Due to the influence of factors such as noise and the working status of the sensor device, the target aspect ratio feature is abnormal, and the BPA of the aspect ratio can be expressed as:

$$E_G : m_G(\text{Br}) = 0, \ m_G(\text{Fr}) = 0.0340, \ m_G(\text{Hr}) = 0.9658,$$
$$m_G(\text{AGM}) = 0.0001, m_G(\text{TBM}) = 0, m_G(\text{X}) = 0.$$

The aspect ratio has a high degree of support for target Hr, while the support degree for Br is 0. Therefore, $E_G$ shows significant conflict with the other evidence. Tables 10 and 11 compare the target recognition performance of the algorithms.

**Table 10.** Comparison of algorithms with $E_F$ and $E_G$ in an error-free environment.

| Method | m(Br) | m(Fr) | m(Hr) | m(AGM) | m(TBM) | m(X) | Target |
|---|---|---|---|---|---|---|---|
| D-S | 0 | 1 | 0 | 0 | 0 | 0 | Fr |
| Yager | 0 | 0.0012 | 0 | 0 | 0 | 0.9988 | X |
| Murphy | 0.7060 | 0.0631 | 0.2003 | 0.0035 | 0 | 0.0270 | Br |
| MSDF | 0.7746 | 0.0702 | 0.1257 | 0.0037 | 0 | 0.0258 | Br |
| Kaur | 0.7945 | 0.0642 | 0.1254 | 0.0034 | 0 | 0.0125 | Br |
| Hu | 0.8563 | 0.0281 | 0.1043 | 0.0021 | 0 | 0.0092 | Br |
| MTS-IFRM | 0.9639 | 0.0345 | $1.23 \times 10^{-4}$ | 0 | 0 | 0 | Br |

**Table 11.** Comparison of algorithms with **E** in an error-prone environment.

| Method | m(Br) | m(Fr) | m(Hr) | m(AGM) | m(TBM) | m(X) | Target |
|---|---|---|---|---|---|---|---|
| D-S | 0 | 1 | 0 | 0 | 0 | 0 | Fr |
| Yager | 0 | 0 | 0 | 0 | 0 | 1 | X |
| Murphy | 0.9830 | $6.31 \times 10^{-4}$ | 0.0163 | 0 | 0 | 0 | Br |
| MSDF | 0.9965 | $5.89 \times 10^{-4}$ | 0.0029 | 0 | 0 | 0 | Br |
| Kaur | 0.9905 | 0.0084 | 0.0011 | 0 | 0 | 0 | Br |
| Hu | 0.9942 | 0.0049 | 0.0009 | 0 | 0 | 0 | Br |
| MTS-IFRM | 0.9811 | 0.0184 | 0.0019 | 0 | 0 | 0 | Br |

Tables 10 and 11 show that because of the conflicting evidence $E_G$, D-S finally determines that Fr is the final result, which is counter-intuitionistic. Meanwhile, the Yager is also unable to correctly recognize the target because it assigns the high-conflict part of the evidence to X. Murphy, MSDF, Kaur, Hu, and the MTS-IFRM can process the conflicting evidence and realize reasonable results. The Murphy method has lower convergence because it calculates the averages without considering the correlations between the evidence, the MSDF method modifies the entropy method to calculate the weight of the evidence, and the Kaur and Hu methods comprehensively improve the credibility of evidence by analyzing the discrepancy in different aspects. Moreover, the accuracy of the MTS-IFRM is higher compared to other methods in the case of fewer features. The MTS-IFRM establishes a higher stability and reliability structure when confronting uncertainty.

The reasons why the MTS-IFRM shows better performance for aerial target recognition can be explained as follows. First, the MTS-IFRM is constructed according to intuitionistic fuzzy theory, which deals with uncertainty data of aerial targets using DPSO-IFCM clustering. Second, the adaptive weight algorithm is used to further improve the classification accuracy of the model, which is crucial for addressing the target recognition problem in an error-free or error-prone environment.

To further verify the effectiveness of the method, a dataset of 10,000 target features is randomly generated within the range given in Table 12 as the test dataset of the simulation.

**Table 12.** Range of the test dataset.

| | Br\|$\delta$ | Fr\|$\delta$ | Hr\|$\delta$ | AGM\|$\delta$ | TBM\|$\delta$ |
|---|---|---|---|---|---|
| FH (km) | 30\|15 | 10\|5 | 2\|1 | 4.5\|2 | 70\|30 |
| DD (km) | 400\|200 | 300\|150 | 150\|75 | 120\|60 | 150\|75 |
| FS (m/s) | 400\|200 | 600\|300 | 100\|50 | 1200\|600 | 2000\|1000 |
| A(m/s$^2$) | 10\|10 | 25\|25 | 15\|15 | 200\|100 | 300\|150 |
| VS(m/s) | 25\|25 | 100\|100 | 20\|20 | 1000\|500 | 2000\|1000 |
| CA (m$^2$) | 0.30\|0.15 | 0.20\|0.1 | 0.10\|0.05 | 0.05\|0.02 | 0.10\|0.05 |
| AR | 1.5\|0.75 | 2.5\|1.0 | 4.0\|2.0 | 8.0\|4.0 | 10.0\|5.0 |

The data model for the simulation feature parameters is:

$$F_{ij} = f_{ij} \pm randn \times \delta_{ij} \tag{39}$$

where $f_{ij}$ denotes the $j$-th feature of the target $i$ corresponding to the deviation $\delta_{ij}$, *randn* denotes a normal random number with a mean of 0 and a variance of 1. Six algorithms with higher recognition rate methods are employed in the experiment.

In Table 13, a( ) represents the recognition rate of the target "·", which is obtained by dividing the number of correctly recognized samples by the total number of testing samples, and in bold is the best simulation result under the same conditions. After fusing the seven features, Figure 7 shows the final recognition rates of six algorithms.

**Table 13.** Recognition rates for five algorithms.

|  | $E_A, E_B$ | $E_C, E_D, E_E$ | $E_F, E_G$ | $E$ |
|---|---|---|---|---|
| D-S | a(Br) = 0.4970 | a(Br) = 0.6085 | a(Br) = 0.7044 | a(Br) = 0.8381 |
|  | a(Fr) = 0.6394 | a(Fr) = 0.8698 | a(Fr) = 0.3802 | a(Fr) = 0.9347 |
|  | a(Hr) = 0.6663 | a(Hr) = 0.9133 | a(Hr) = 0.5227 | a(Hr) = 0.9997 |
|  | a(AGM) = 0.6102 | a(AGM) = 0.8389 | a(AGM) = 0.4165 | a(AGM) = 0.9160 |
|  | a(TBM) = 0.4207 | a(TBM) = 0.7138 | a(TBM) = 0.2678 | a(TBM) = 0.9041 |
| Murphy | a(Br) = 0.5976 | a(Br) = 0.7861 | a(Br) = 0.5915 | a(Br) = 0.9174 |
|  | a(Fr) = 0.7350 | a(Fr) = 0.8473 | a(Fr) = 0.7326 | a(Fr) = 0.9064 |
|  | a(Hr) = 0.7956 | a(Hr) = 0.9602 | a(Hr) = 0.7904 | a(Hr) = 0.9990 |
|  | a(AGM) = 0.6463 | a(AGM) = 0.7983 | a(AGM) = 0.6228 | a(AGM) = 0.9044 |
|  | a(TBM) = 0.4862 | a(TBM) = 0.6907 | a(TBM) = 0.4968 | a(TBM) = 0.8611 |
| MSDF | a(Br) = 0.6173 | a(Br) = 0.7892 | a(Br) = 0.6141 | a(Br) = 0.9087 |
|  | a(Fr) = 0.7709 | a(Fr) = 0.8557 | a(Fr) = 0.7776 | a(Fr) = 0.8951 |
|  | a(Hr) = 0.8553 | a(Hr) = 0.9749 | a(Hr) = 0.8489 | a(Hr) = 0.9990 |
|  | a(AGM) = 0.6977 | a(AGM) = 0.8217 | a(AGM) = 0.7023 | a(AGM) = 0.9212 |
|  | a(TBM) = 0.5365 | a(TBM) = 0.7119 | a(TBM) = 0.5362 | a(TBM) = 0.8549 |
| Kaur | a(Br) = 0.6215 | a(Br) = 0.8021 | a(Br) = 0.6042 | a(Br) = 0.9213 |
|  | a(Fr) = 0.7821 | a(Fr) = 0.8566 | a(Fr) = 0.7511 | a(Fr) = 0.9155 |
|  | a(Hr) = 0.8163 | a(Hr) = 0.9713 | a(Hr) = 0.8224 | a(Hr) = 0.9990 |
|  | a(AGM) = 0.7062 | a(AGM) = 0.8078 | a(AGM) = 0.6634 | a(AGM) = 0.9156 |
|  | a(TBM) = 0.6035 | a(TBM) = 0.7256 | a(TBM) = 0.5264 | a(TBM) = 0.8744 |
| Hu | a(Br) = 0.7654 | a(Br) = 0.8156 | a(Br) = 0.6317 | a(Br) = 0.9315 |
|  | a(Fr) = 0.7905 | a(Fr) = 0.8557 | a(Fr) = 0.7812 | a(Fr) = 0.9213 |
|  | a(Hr) = 0.8632 | a(Hr) = 0.9812 | a(Hr) = 0.8497 | a(Hr) = 0.9992 |
|  | a(AGM) = 0.7256 | a(AGM) = 0.8247 | a(AGM) = 0.7123 | a(AGM) = 0.9336 |
|  | a(TBM) = 0.6636 | a(TBM) = 0.7311 | a(TBM) = 0.5546 | a(TBM) = 0.8639 |
| MTS-IFRM | a(Br) = 0.8834 | a(Br) = 0.7145 | a(Br) = 0.8345 | a(Br) = 0.9354 |
|  | a(Fr) = 0.7341 | a(Fr) = 0.8844 | a(Fr) = 0.5341 | a(Fr) = 0.9555 |
|  | a(Hr) = 0.7589 | a(Hr) = 0.9253 | a(Hr) = 0.8835 | a(Hr) = 0.9952 |
|  | a(AGM) = 0.7954 | a(AGM) = 0.9051 | a(AGM) = 0.4954 | a(AGM) = 0.9862 |
|  | a(TBM) = 0.8795 | a(TBM) = 0.9493 | a(TBM) = 0.7101 | a(TBM) = 0.9899 |

In Figure 7, the MTS-IFRM algorithm has better performance than the other five methods and is slightly inferior to other algorithms for the Hr. The main reasons for this: in other methods, the preliminary recognition of the target with the fuzzy membership function will have high accuracy, and the results will be fused by the evidence theory method. Moreover, Table 2 shows that the features of flight height and speed for Hr have a large difference from those of other targets, for example, suppose the radar detects a suspicious target, the target features are: $A = 1.7$ km, $B = 135$ km, $C = 75$ m/s, $D = 10$ m/s$^2$, $E = 25$ m/s, $F = 0.09$ m$^2$ and $G = 3.8$, in our proposed method, the existing target features are used to construct the T-S intuitionistic fuzzy training model, and the feature datasets and model parameters of the target are updated with the recognition result, which has higher requirements for training data. If the number of Hr in the training data is insufficient, the suspicious target may be recognized as an AGM or TBM with $E_A$ and $E_B$, because Hr, AGM, and TBM have the similar feature ranges of detection distance. In addition, due to the similar feature ranges of acceleration and vertical speed, the suspicious target may be recognized as a Br or Fr with $E_C$, $E_D$ and $E_E$. However, the final recognition rate of the MTS-IFRM is more than 99% for the Hr with abundant training datasets. Overall, if a

richer and more effective training dataset can be obtained, the recognition accuracy of the proposed MTS-IFRM can be improved.

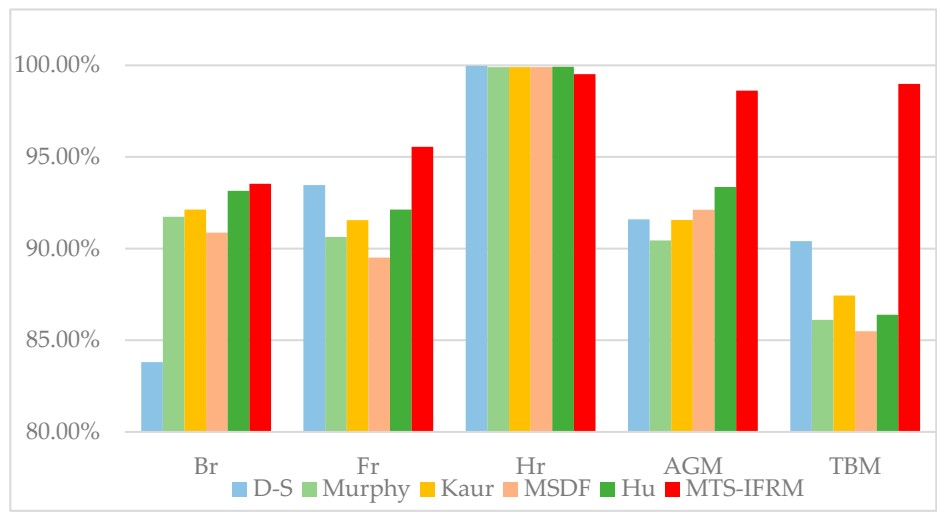

**Figure 7.** The recognition rates of six algorithms.

## 5. Conclusions

In this paper, a target recognition approach based on MTS-IFRM is proposed, which constructs a fuzzy classification model to enhance the robustness of the recognition process. The intuitionistic fuzzy theory and ridge regression method are employed in the consequent identification, the intuitionistic fuzzy C-regression clustering based on dynamic optimization can realize the premise identification. Then, the adaptive weight algorithm improves the classification accuracy of the corresponding model. The experimental results show that the MTS-IFRM can effectively recognize aerial targets in error-free and error-prone environments, and its performance is better than the methods proposed for aerial target recognition.

Although the proposed MTS-IFRM can show encouraging results for target recognition, many issues remain. For example, when fusing the outputs of multiple models, the method of the weight distribution is still relatively rough. As the features of the target increase, a more complete weight allocation algorithm needs to fuse the outputs of multiple models accurately. In the future, further methods can be proposed to improve accuracy by extending the models to adjust to different types of datasets and by developing more efficient objective functions for the MTS-IFRM using specific samples.

**Author Contributions:** Conceptualization, C.Z., W.X. and Z.L.; methodology, C.Z., W.X. and Z.L.; software, Y.L. and Z.L.; formal analysis, C.Z., Y.L. and Z.L.; investigation, Y.L. and Z.L. resources, Y.L. and Z.L.; data curation, C.Z.; writing—original draft preparation, C.Z.; writing—review and editing, C.Z., W.X. and Z.L.; visualization, C.Z.; supervision, C.Z.; project administration, C.Z.; funding acquisition, Z.L. All authors of the article have provided substantive comments. All authors have read and agreed to the published version of the manuscript.

**Funding:** This work was supported by the National Natural Science Foundation of China (62171287, 62076165), the Science & Technology Program of Shenzhen (No.JCYJ20220818100004008), the Innovation Team Project of the Department of Education of Guangdong Province (No. 2020KCXTD004) and the Science and Technology on Information Systems Engineering Laboratory (No. 05202206).

**Data Availability Statement:** The data presented in this study are partly available on request from the corresponding author. The data are not publicly available due to their current restricted access.

**Conflicts of Interest:** The authors declare no conflict of interest.

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
