# Peer review of "Multi-Source T-S Target Recognition via an Intuitionistic Fuzzy Method"

_remotesensing, doi:10.3390/rs15245773_

Round 1

Reviewer 1 Report

Comments and Suggestions for Authors

Generally, this paper is well written and is technically sound. But it still requires minor revision before publication. My comments are as follows:

(1)The complexity analysis of the new algorithm should be added.

(2)Please explain what the relationship between the GPSO-IFCM clustering algorithm and the adaptive weight algorithm.

Comments on the Quality of English Language

Overall satisfactory but minor editing of English language is required.

Reviewer 2 Report

Comments and Suggestions for Authors

This paper presents a multi-source Takagi–Sugeno (T-S) intuitionistic fuzzy rules method (MTS-IFRM) for aerial target recognition in complex environments. The paper is well written and well organised and introduces an interesting research work.  Below are some comments that might be helpful to improve the quality of the paper:

1-Authors need to revise the paper to ensure enhancing its quality of writing. For example in Line 116 - Line 135: The sentence "The classification and target recognition process are shown in Figure.1." could be revised to "Figure 1 illustrates the classification and target recognition process.". Also, please check for minor typos and grammatical inconsistencies throughout the document. For example, there are typographical issues like "Figure.4" and "Figure. 5" – consistency in formatting is important.
2-There seems to be an inconsistent use of mathematical notations. Check for this and ensure that consistent notations should be maintained throughout the document.
3-Terminology like "premise parameter of CS and AR features", "intuitionistic fuzzy sets", "sequencing and decision-making" appear without sufficient background or description, which might alienate some readers. 
4-The paper introduces many parameters like w, c1, c2, T, t, wmax, and wmin. There needs to be a tabulated section at the beginning or the end of the section which concisely defines each parameter for quick reference.
5-While the mathematical formulation provides the essence of the algorithm, the paper would benefit from more intuitive explanations. For example, a simple visualization or flowchart showing how DPSO adapts might help to improve the quality of presentation for the paper (similarly, the role of inertia weight and its dynamic adjustment could be better illustrated with a graphical representation.). 
6-There are instances where the same concept might be referred to in slightly varying terms, e.g., "current best position" and "optimal position". This can cause confusion.
7-Some of the result tables are very long with insufficient explanation of results in the main body of the manuscript. Authors can consider more summarisation of tabular results and add more graphs with better result discussion. For example, Tables 1 and 2 are dense with information. Some additional explanation or simplification may be beneficial for readers less familiar with the subject.
8-Better in text description are required for the figures. For example, Figure descriptions are necessary. "Figure 4 shows the fuzzy membership..." - but what specifically should the reader look for or conclude from this figure?
9-The reason behind selecting the mentioned algorithms (D-S, Yager, etc.) for comparison is not clear. A brief on why these particular algorithms were chosen would be helpful.

Comments on the Quality of English Language

The paper requires a thorough language review.

Reviewer 3 Report

Comments and Suggestions for Authors

In this manuscript, the authors proposed multi-source Takagi–Sugeno (T-S) intuitionistic fuzzy rules method to recognize aerial targets. I have some questions:

1. In my opinions, the authors should enhance why they have to use "fuzzy" method to recognize aerial targets. Is there any reason that fuzzy has to be used in aerial target recognition? The authros have to address this.

2. The compared methods are too old, [19],[29],[30],[31]. Are there any reasons to compare the proposed method with these references? Why not compare with some newer ones.

3. I suggest authors could use more figures for demonstration. For example, I would like to see some of the datasets.

4. Is it possible to show some mis-recognized cases and provide some analysis if possible. 

Comments on the Quality of English Language

In my opinion, the English is generally well written

Round 2

Reviewer 3 Report

Comments and Suggestions for Authors

After carefully reviewing, I can see that the authors have addressed all my questions, and I have no other extended questions.

Comments on the Quality of English Language

In my opinion, English is well.